# Untangling the Small Intestine in 3D cine-MRI using Deep Stochastic Tracking

**Louis D. van Harten**[1]                                    L.D.VANHARTEN@AMSTERDAMUMC.NL
**Catharina S. de Jonge**[2]
**Jaap Stoker**[2]
**Ivana Išgum**[1,3]

[1] *Department of Biomedical Engineering and Physics, Amsterdam UMC, Location AMC, University of Amsterdam, Amsterdam, The Netherlands.*

[2] *Department of Radiology and Nuclear Medicine, Amsterdam Gastroenterology, Endocrinology and Metabolism, Amsterdam UMC, University of Amsterdam, Amsterdam, The Netherlands.*

[3] *Department of Radiology and Nuclear Medicine, Amsterdam UMC, Location AMC, University of Amsterdam, Amsterdam, The Netherlands.*

## Abstract

Motility of the small intestine is a valuable metric in the evaluation of gastrointestinal disorders. Cine-MRI of the abdomen is a non-invasive imaging technique allowing evaluation of this motility. While 2D cine-MR imaging is increasingly used for this purpose in both clinical practice and in research settings, the potential of 3D cine-MR imaging has been largely underexplored. In the absence of image analysis tools enabling investigation of the intestines as 3D structures, the assessment of motility in 3D cine-images is generally limited to the evaluation of movement in separate 2D slices. Hence, to obtain an untangled representation of the small intestine in 3D cine-MRI, we propose a method to extract a centerline of the intestine, thereby allowing easier (visual) assessment by human observers, as well as providing a possible starting point for automatic analysis methods quantifying peristaltic bowel movement along intestinal segments. The proposed method automatically tracks individual sections of the small intestine in 3D space, using a stochastic tracker built on top of a CNN-based orientation classifier. We show that the proposed method outperforms a non-stochastic iterative tracking approach.

**Keywords:** Deep learning, cine-MRI, centerline extraction, small intestine, motility.

## 1. Introduction

Intestinal motility consists of contractions of the bowel wall, providing peristaltic movement and intestinal content mixing. Changes in small bowel motility are associated with a wide variety of functional gastrointestinal diseases and disorders such as Crohn's disease and chronic intestinal pseudo-obstruction (CIPO) (Menys et al., 2018; Paine et al., 2013; van Rijn et al., 2020). While motility in the proximal small intestine can be measured invasively using antroduodenal manometry, the high patient burden of this procedure has driven the investigation of cine-MRI as a non-invasive alternative for the complete small bowel. The resulting images can be visually assessed by a radiologist (Guglielmo et al., 2015; Heye et al., 2012), or quantitatively evaluated using techniques like displacement mapping (Odille

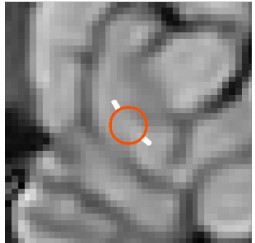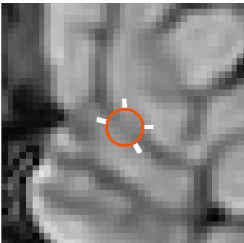

Figure 1: A bowel section with two unambiguous directions (left) and an ambiguous case where the classifier may predict directions for different intestinal sections (right).

et al., 2012), or diameter measurements (Wakamiya et al., 2011). While diameter measurements traditionally involve labour-intensive annotations, recent works have shown promise in automating this task using convolutional neural networks (CNNs) (Wu et al., 2020).

Currently, literature on quantifying intestinal motility in cine-MRI is generally centered around 2D cine-MRI (De Jonge et al., 2018). Although some works acquire 3D cine-MRI, the analysis is generally limited to sequences of 2D slices (Menys et al., 2018; Odille et al., 2012; Wakamiya et al., 2011; Wu et al., 2020). As movement of the small intestines occurs continuously in all directions, motility movement and through-plane motion are entangled in such methods: in 2D analysis it is impossible to differentiate intestinal contractions from bowel sections moving (partially) in and out of plane, as both lead to a change in perceived luminal diameter between time points. Furthermore, 2D analysis is ill suited for differentiating peristaltic from intestinal content mixing motion and for identification of dysfunctional propagation of contractions, as doing so requires correlating contractions along intestinal segments.

The low-dimensional analysis can be largely attributed to the lack of (semi-)automatic image analysis tools enabling investigation of the intestines as 3D structures. A stretched-out, untangled representation of the intestines would allow easier assessment by human observers and it could enable detailed automatic analysis of the functional peristaltic motion. Such a representation can be generated by piecewise resampling of the intestines along their centerline, creating a multi-planar reformation (MPR), but manually annotating such centerlines is unfeasible in routine clinical patient work-up. Previous work on automatic extraction of centerlines from the small intestine in MR has relied on the availability of an accurate segmentation mask of the small intestine (Spuhler et al., 2006). However, recent work on deep learning-based segmentation has shown that state-of-the-art methods achieve much lower performance in the small intestine compared to other organs, producing clinically acceptable segmentations in less than 40% of the tested subjects (Liu et al., 2020). The authors indicate that the large diversity in shapes, sizes and locations in the abdomen among different patients make the small intestine especially difficult to segment using deep learning. This means the extraction of centerlines through segmentation is currently not a promising approach for application in the clinic. To the best of our knowledge, more recent work on the extraction of centerlines in the small intestine is not available.

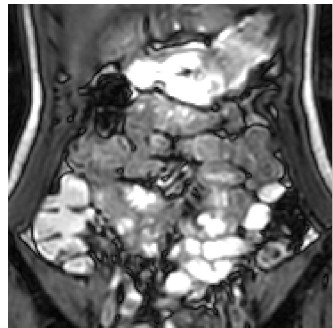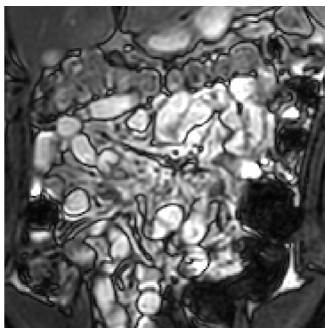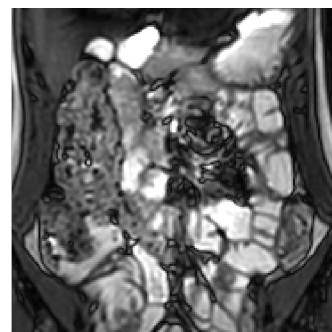

Figure 2: Center-cropped slices from three subjects, illustrating variety in characteristics.

Alternative to using segmentation maps as an intermediate, centerlines can also be extracted directly from the MR images. Existing methods for centerline extraction in medical images have been developed for application in vessels in various anatomical regions, as well as the airways in chest images (Friman et al., 2010; Bauer et al., 2009). These methods have traditionally relied on modelling the target anatomy as tubular structures (Frangi et al., 1998). More recent work has used machine learning approaches to predict centerlines either directly (Sironi et al., 2015) or through use of intermediate flow fields (Gülsün et al., 2016). Recent work in cardiac CTA has shown the efficacy of using deep neural orientation classifiers for extracting centerlines in the coronary artery tree (Wolterink et al., 2019). Inspired by this work, we develop a tracking method for the small intestine.

In abdominal 3D cine-MRI, substantial sections of the small intestine are outside of the field of view (FOV), meaning the centerline has to be extracted as unconnected segments. The small FOV is a consequence of the short time available to scan each time point. Limited image context is available and there is a possibility of the tracker crossing the intestinal wall into an adjacent bowel segment (Figure 1), incorrectly fusing unconnected segments. To overcome these challenges, we introduce a novel stochastic tracking strategy, which improves the robustness of the tracking compared to non-stochastic neural centerline extraction.

## 2. Data

The method was developed and evaluated with 3D cine-MR scans from fourteen healthy volunteers, retrospectively selected from a previous study (de Jonge et al., 2019). Sequences were imaged at 1.0 image per second during a 20 second breath-hold, acquired at 2.5x2.5x2.5 mm and reconstructed to 1.4x1.4x2.5 mm. Image FOV is 400x400x35 mm. The exact anterior-posterior planning position differs, but all scans contain the terminal ileum.

Prior to scanning, subjects were orally administered 1000 mL mannitol solution to improve MR contrast in the intestine. The images were acquired using a balanced fast field echo sequence, aimed at maximising the contrast of the interface between the intestinal wall and the surrounding fatty tissue by means of anti-phase annihilation. Due to the sensitivity to body composition of this effect, the images present a large variability in contrast. Furthermore, contrast at the interface of adjacent intestinal sections without fatty tissue in between them varies strongly among subjects (Figure 2).

The reference centerlines were annotated as segments by manually placing markers along the direction of the small intestine until the intestine would leave the FOV, or until the annotator could no longer distinguish the direction of the segment due to image artifacts or low contrast. Annotations were made in one of the 3D time points for each subject. An average of 1150 mm of reference centerlines were annotated per subject, divided over 186 segments of on average 88 mm. Segments shorter than 25 mm were excluded, resulting in a total of 181 annotated segments in the reference set.

## 3. Methods

### 3.1. Deep neural centerline tracking

Building on a method for coronary artery tracking in CTA (Wolterink et al., 2019), we develop a method to track centerlines through the small intestine. The method uses a deep neural network to predict direction proposals. This orientation classifier consists of a fully convolutional neural network that operates on 32x32x32 voxel patches from a single 3D cine-MR time point. The output is a set of 500 logits that correspond to equidistant vectors on the unit sphere. During training, the network is provided with patches sampled around a centerline and is trained to predict a probability distribution for its directions. Once the classifier is trained, a centerline can be tracked by starting at a seed point and iteratively stepping into the direction with maximum probability $p_{max}$. By masking the probabilities for backward directions, this allows stepping through an estimate of the centerline until a stopping criterion is met. If a moving average of the maximum value of the probability distribution drops below a threshold, the tracker is terminated. For each starting point, the method produces two centerlines (one in each direction), which are fused by reversing the direction of one of them and concatenating the two segments.

### 3.2. Stochastic tracking strategy

To improve the robustness and reliability of the tracking system, we propose the concept of a stochastic agent. Stochastic agents behave similarly to a non-stochastic iterative tracker, but these agents do not always step into the direction with the maximum probability. Instead, they stochastically sample a direction on the unit sphere from the generated probability distribution (i.e. a direction with estimated probability $p$ will have a $p$ chance of being chosen by the agent). By initialising multiple such agents, the probability space for possible centerline paths can be explored.

For the stochastic tracker, we initialise $n$ stochastic agents at random points in a sphere with radius $r$ around the seed point. All agents start tracking simultaneously. To combine these agents into a single centerline, the median location of all living agents is computed after each step. If an agent strays more than distance $d$ away from the median location, the aberrant agent is terminated; if more than $t$ agents are terminated, tracking stops. By enforcing the distance-to-median constraint, the system effectively tracks by majority vote: decisions to move into a direction are only taken if the majority of agents agree. Agents are not re-initialized throughout; the progress of the agents is independent from the predicted centerline points apart from the aberrant termination condition.

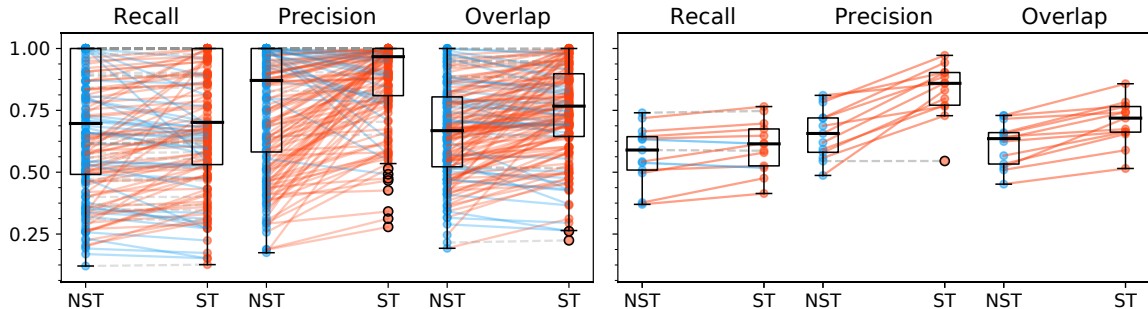

Figure 3: Comparison of results achieved by non-stochastic tracking (NST, blue) and stochastic tracking (ST, red) in terms of recall, precision and overlap. Left: Each point represents an intestinal segment. Right: Each point represents a subject. Lines connect results for the same segments/subjects in both experiments, their colour indicates which strategy performed better on each metric. When both methods achieve similar performance (metric difference < 0.01), points are connected by a dashed grey line.

### 3.3. Evaluation

The centerline segments produced by the non-stochastic and the stochastic method are evaluated based on their overlap with the reference, following the definition in (Schaap et al., 2009): The harmonic mean between recall and precision, similar to the Dice coefficient. As radius annotations are not available, we use a static distance threshold of 10 mm to define true positive, true negative, false positive and false negative points (TP/TN/FP/FN). Due to the incomplete reference standard, precision may be underestimated in some intestinal segments. Beside evaluating the results per segment, we analyse the results per subject by pooling all TP/TN/FP/FN.

## 4. Experiments and results

### 4.1. Experimental settings

The reference centerlines were quantized at a resolution of 0.5 mm to uniformly generate training points along the segments. During training, patches were sampled at an isometric resolution of 1.5 mm around the training points using linear interpolation, resulting in a receptive field of 48 mm. Full 3D rotation augmentations were used. Due to the low number of subjects in this study, our experiments were performed with leave-one-subject-out cross-validation. The networks were trained using a cosine annealed learning rate policy with warm restarts (Loshchilov and Hutter, 2016). Suitable values for maximum learning rate, number of epochs and cut-off values for the stopping criteria were selected based on preliminary experiments on a single image. This image was excluded from the evaluation. Tracking step size was set to 0.5 mm and confidence-based stopping criterion thresholds were set to $p_{max} > 0.025$ with a moving average window of 3 steps for both tracking methods.

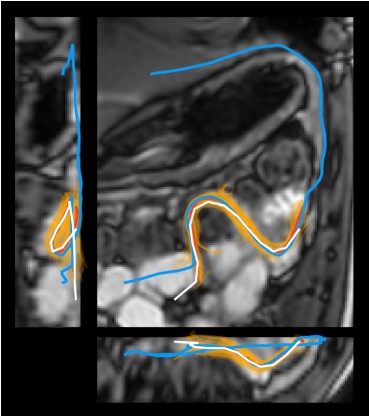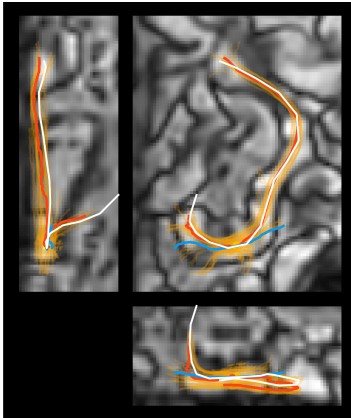

Figure 4: Visualisation of tracking results for an intestinal segment in two different subjects. Triplar view centered on the point where the trackers were initialized and zoomed to fit the results to the view. Projections are plotted for the manual reference (white), the non-stochastic tracking result (blue), the stochastic tracking result (red) and the individual stochastic agents (orange).

For the stochastic tracker, we used $n = 64$ agents, initialized in a sphere of $r = 5$ mm around the seed point, equivalent to the thinnest non-contracted regions of the intestinal lumen. The maximum median-distance for individual agents was set to $d = 10$ mm and the stopping criterion threshold for the maximum number of terminated agents was set to $t = \frac{3}{4}n$. During evaluation, a single seed point on the reference centerline was randomly selected in each intestinal segment, serving as a starting point for the automatic methods.

### 4.2. Results

The quantitative results for both methods are listed in Table 1 and shown in Figure 3.

Table 1: Quantitative results for the non-stochastic and stochastic tracking methods.

| Strategy | Segments | | | Subjects | | |
|---|---|---|---|---|---|---|
| | recall | precision | overlap | recall | precision | overlap |
| Non-stochastic | 0.69 | 0.77 | 0.67 | 0.57 | 0.66 | 0.61 |
| Stochastic | 0.72* | 0.88* | 0.75* | 0.61* | 0.83* | 0.69* |

*statistically significant improvement (Wilcoxon signed rank test, p<0.05)

Figure 4 shows a qualitative comparison for two intestinal segments. The images on the left show results for a segment where both methods have near-perfect recall (i.e. follow the reference along its entire length), but the non-stochastic tracker (blue) failed to terminate when encountering a visually ambiguous area. This caused it to "leak" out of the intestine, tracking a loop around the stomach. On the other end of the segment, it crosses the

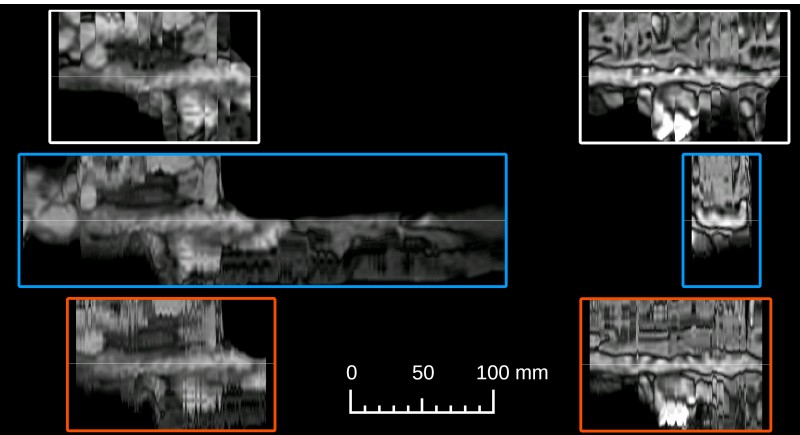

Figure 5: Untangled representations of the intestinal segments shown in Figure 4, generated from the manual centerline (top), the non-stochastic tracking result (middle) and the stochastic tracking result (bottom), aligned along the horizontal axis. Deviant segment lengths are caused by low precision and low recall, respectively.

intestinal wall into an adjacent bowel loop. In both cases, at least one stochastic agent agreed with the non-stochastic method, but was terminated as aberrant. The images on the right show a segment where both the non-stochastic and the stochastic method have near-perfect precision, but the non-stochastic method fails due to collision with the intestinal wall in both directions. This is caused by a region in the segment where orientation classification performance is poor. While the stochastic agents have the same problem, a large cluster of them survives the problematic area, causing the complete section to be tracked correctly. Additional qualitative results can be found in the appendix.

Finally, Figure 5 shows the untangled representations for both of these intestinal segments, generated from both the manual and the automatic centerlines. Rotation angle around the centerline and horizontal axis position were matched for easier visual comparison. Differences in length are caused by late and early termination of the non-stochastic method on these two segments. In the top-left view, the intestine exhibits a slight wobble around the centerline, revealing an imperfection in the manual annotation. In the correctly tracked section, both methods produce a more straight result. The representations produced by the stochastic method contain more high-frequency resampling noise in the outer regions of the MPR, indicating a lower smoothness of the centerline.

## 5. Discussion

We have presented a novel stochastic method for tracking centerlines through the small intestine in 3D cine-MRI. To the best of our knowledge, this is the first method for tracking centerlines in the small intestine that does not depend on the availability of an accurate segmentation mask. The method is inspired by a recent method that accurately extracted centerlines of the coronary arteries in cardiac CTA using a deep learning-based iterative

tracker. We have presented measurable improvement by adding stochasticity: centerlines produced by our method are less prone to crossing the intestinal wall and leaking out into surrounding tissues. Our method is fully parallelizable, meaning no performance penalty is incurred by the stochastic strategy if sufficient computation cores are available.

Quantitatively, the stochastic method outperformed the non-stochastic method in terms of precision and overlap scores. While the improvement in recall was also statistically significant on both patient and segment levels, the difference was much smaller than for the other two metrics. The reason is that the stochastic method was more likely than the non-stochastic method to terminate early in regions where network performance is compromised, for example due to the presence of artifacts. It may be beneficial to relax the thresholds on the stopping criteria, trading off improvements in precision for additional gains in recall. Results of the non-stochastic method shown in Figure 5 illustrate that for visual assessment, low precision is not as problematic as low recall. Qualitatively, the MPRs generated from the results of the stochastic method look noticeably different from MPRs generated from the non-stochastic results. The reason for this is lower smoothness of the centerline: when a stochastic agent is terminated, the median tracking location jumps away from the direction of the dying agent. Should this pose a problem for downstream tasks, the noise could be removed by applying a smoothing filter to the extracted centerline.

Performance of our method may be affected by the noisy reference annotations. They were created by a single annotator and because of the difficulty of the task, this likely resulted in a number of inaccuracies. Furthermore, due to the annotation protocol dictating human uncertainty as a stopping criterion, the annotations are biased to avoid difficult decisions. For this reason, precision may have been underestimated in evaluation for some intestinal segments, caused by incomplete reference centerlines. Future work will employ multi-observer consensus to alleviate these issues. Furthermore, unlike in work proposed by (Wolterink et al., 2019), our reference annotations did not define radii around the centerline, preventing the methods from using variable step-sizes and orientation correcting off-centerline translation augmentations. Prior work has shown such augmentations can result in substantial performance improvements. Future work could focus on acquiring radius annotations in the small intestine, or develop a functionally similar augmentation strategy that circumvents the need for radius annotations.

While the method was developed for a 4D modality, it only operates on one time point in the sequence. Hence, it does not exploit the available 4D information. Future work could investigate methods to augment the orientation classifier with 4D input patches, or to employ tracking agents in multiple time points to further improve robustness.

## 6. Conclusion

In this work, we have demonstrated the feasibility of automatic centerline tracking through the small intestine in 3D cine-MR images using deep neural trackers. We have presented a novel stochastic tracking strategy, which improves tracking robustness by exploiting a multi-agent consensus. The presented method outperforms non-stochastic iterative tracking across all of the used evaluation metrics. Automatic untangling of the small intestine paves the way to automatic motility analysis in 4D.

## Acknowledgments

The authors would like to thank Jelmer Wolterink for his valuable input at the early stages of this project.

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

## Appendix A. Additional visual examples

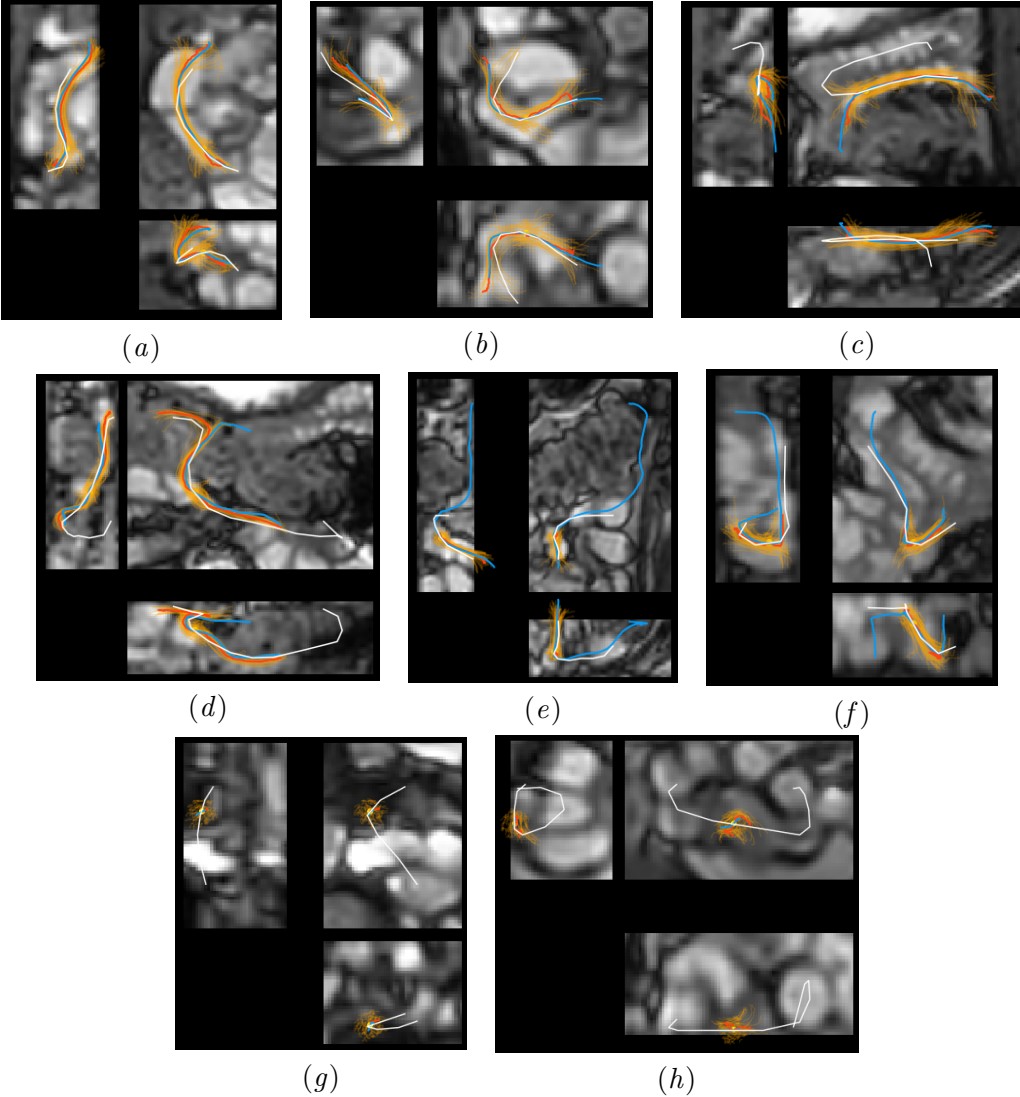

$(a)$             $(b)$             $(c)$

$(d)$             $(e)$             $(f)$

$(g)$             $(h)$

Figure 6: Additional qualitative examples, showing various situations. Projections are plotted for the manual reference (white), non-stochastic tracking results (blue), stochastic tracking results (red) and individual stochastic agents (orange). (a-c) Both methods perform similarly. (d) Result from non-stochastic tracker crosses the intestinal wall. (e-f) Cases with the largest decrease in recall: Stochastic tracker terminates early as more than $t$ agents hit the confidence-based and aberrant stopping criteria. (g-h) Seed points inside an air bubble and near the FOV border, compromising network performance.

