# OpenReview forum: "Untangling the Small Intestine in 3D cine-MRI using Deep Stochastic Tracking"
_MIDL.io/2021/Conference — MIDL 2021_

### Official Review · ~Hans_Meine1 · 2021-03-04

**Confidence:** 5
**Preliminary Rating:** 4
**Recommendation:** Oral, Poster
**Final Rating:** 4

**Summary:**

The authors build upon a recent CNN-based 3D centerline tracking method developed for coronary arteries (by the same group) and make two contributions: On the one hand, they apply it to a dataset of 3D cine MRI of the small intestines, which is a relatively new and unexplored medical image analysis domain and thus contributes as a validation / application contribution, albeit on a relatively small dataset (14 subjects). Second, they make a small but significant methodological contribution by introducing a number of stochastic trackers and some kind of majority voting, which helps to alleviate problems with this particular dataset.

**Strengths:**

The paper is well-written and nicely illustrated, follows the standard structure, adequately summarizes relevant prior work and also discusses its own contributions well.

I found the method section lacking, in particular the introduction of the deep neural centerline tracking which this paper is centrally based on, but that was probably for lack of space. Maybe it would help to explicitly point the reader to the prior work which describes this in much more detail; the sentence "inspired by a method …" does not sound like one would find all necessary algorithmic detail in the cited paper.

The method is definitely well-suited for this task, and likely superior to previous works on similar images. I am not sure how (un)common 4D MRI is for the small intestines – the imaging itself may be quite a challenge here. In particular, the motivation for directly tracking the centerline instead of starting with a segmentation mask is sound.

Fig. 4 was particularly helpful in understanding how the stochastic tracking works in practice.

**Weaknesses:**

One weakness is that the dataset is relatively small; the authors have only 14 MRI datasets from healthy volunteers. Therefore, the evaluation also does not perform a proper training / validation / testing split, but a leave-one-out cross validation is used. Given this small dataset, this is certainly a good idea, although I wonder if the one split used for algorithmic development should have been excluded from the evaluation. (The authors state that they re-trained this fold with a different random seed.)

I also think that the surface DSC is a strange choice, given the fact that the reference annotations do not even have radius information. Since the whole evaluation is based on (surface-based) precision and recall measures, a centerline-based variant would seem more adequate and even simpler. In the end, this is not a serious weakness, though – I believe the results would not change much.

**Deanonymize Review:**

yes

**Detailed Comments:**

It would've been nice to visualize the centerlines also in Fig. 2 (and to refer to it at the end of sec. 2). Even if this is admittedly difficult in 2D, it could indicate the coarse location in relation to the full abdominal ROI.

I repeatedly stumbled over the term "intestinal segments", which sounded like a medical term, but apparently referred to the coarse polyline segments from the reference. Maybe this could be just shortened to "segments".

The paper states that "segments shorter than 25mm were excluded", but neither why nor how exactly. (Is one of the endpoints removed from the polyline? Are the two points merged within the polyline, e.g. to the segment center? Or does the whole polyline fall apart into 2 or more pieces?)

In the end, I believe the answer to the previous question(s) is that the segments are only ever considered in isolation, not the full polygonal centerline. This leads to distributions over 181 results instead of 14 (cf. Fig. 3), but I wonder what the meaning of these segments is. Wouldn't either subject-level evaluation, or distributions over a fine, resampled centerline make more sense?

If you rotate the images in Fig. 4, you could enlarge them without additional space.

The paper refers to "the reference surface", although that was never introduced. In fact, the discussion later explicitly states that "our annotations did not define radii around the centerline". See my remarks on the sDSC choice in the "Weaknesses" section.

With respect to the "noise introduced by dying agents", wouldn't it be possible (and sensible) to exclude them over the full lenght?

The text repeatedly talks about "substantial" differences instead of using proper statistical significance. I believe this may be due to the small dataset, but it would be worth stating explicitly why no significance analysis is performed. Qualitatively, the example results look convincing.

The title mentions tracking "in 4D MRI", but only a single timepoint is used. This may be misleading, in particular if people are actively looking for 4D methods by keyword.

I believe 3D cine-MRI is a relatively unknown modality, and I got quite some questions which were not addressed by this manuscript. For instance, "cine" MRI typically refers to data aggregated over many cycles of periodic movement (heart beats), as opposed to simple 4D MRI ("realtime" MRI in case of cardic imaging). In this case, I am not sure how periodic the movement is, and how this can be measured and appropriately aggregated. Of course, such a discussion may be far outside the scope of this paper, yet some hints at the particular challenges or a statement on how (un)common such data is would be welcome.

**Final Rating Justification:**

I already voted for acceptance before the revision, and the authors did a good job with their revision and the replies to my comments and those of the other reviewers, so I am even more convinced now of this contribution. (Just noticed the rating is missing a "t", but I can't change that…)

**Justification Of The Preliminary Rating:**

I would maybe rate this paper 8/10 or so – it is not without weaknesses, but it is not just a "weak accept". It advances the state of the art in segmentation of the intestines and should be interesting to the MIDL audience.

W.r.t. recommendation for the MedIA special issue, I am a little unsure - if the evaluation points would be addressed and additional results are included, I think I would recommend the extended version.

**Paper Type:**

both

**Questions To Address In The Rebuttal:**

The choice of evaluation measures should be justified (or changed), both in terms of surface vs. centerline, as well as in terms of aggregation levels (centerline positions vs. "segments" vs. subjects).

**Special Issue:**

no

---

> ### Author Response · Authors · 2021-03-17
> **response to reviewer 1**
>
> We thank the reviewer for the many helpful suggestions. We address the raised issues below.
>
> Weaknesses:
> We understand the concern regarding the inclusion of the image used in development. We have excluded this image and re-run the result analysis. The manuscript is updated to reflect the numbers from the remaining 13 folds (section 4).
>
> We agree with the reviewer that the choice of evaluation metrics is not ideal. Preferably, we would define true positives as points on the centerline at a shorter distance from the reference than the local radius (i.e. inside of a segmentation mask). This was proposed by Schaap et al. (2009, MedIA) for their standardized overlap measure in scoring centerlines in coronary artery tracking. As we do not have annotated radii in our reference set, we instead use a static maximum distance of 10mm (a typical radius, rather than the true radius) to approximate the overlap. This happens to coincide with the TP/FP/TN/FN definitions used for the surface DSC, if applied to surfaces with zero area (i.e. 3D line segments like centerlines). Here our terminology was indeed confusing, as a surface with zero area is generally considered a curve, not a surface. We have added a reference to Schaap et al. and we changed the evaluation section to match their terminology (section 3.3).
>
> Comments:
> Regarding Figure 2, we prefer not to overlay the centerlines in order to better visualize the differences in image characteristics.
>
> Intestinal segments is indeed not a medical term. These segments are parts of the centerline that could be connected by a human observer; by using the term, we stress that these are derived from the visible anatomy, as opposed to arbitrarily defined for the method.
>
> With the current scanning protocol, it is unfortunately not possible to join the segments into a single polyline: many of the intestinal loops are (partially) out of view due the limitations in the FOV. This means that information allowing to connect visible segments to each other is lacking, even to a manual annotator. The endpoints of the segments are generally at the border of the image. This is why we work on a per-segment basis; the final paragraph of the introduction has been updated to clarify this. Short segments were excluded as these are unlikely to carry enough information to be relevant in downstream tasks. Also, the length of short segments is close to the intestine radius. This means they would all score relatively high on the overlap measures, regardless of the automatic centerline quality. By excluding short segments, we avoid artificially inflating our results.
>
> While we cannot compute the overlap over an uninterrupted polyline, it is possible to pool all of the TP/FP/TN/FN results per patient and perform the evaluation per patient instead of per segment. We have added results of this analysis to the manuscript (section 4.2).
>
> We would prefer not to rotate figure 4. While the rotation would save space, we prefer to keep the coronal view upright. Currently, readers could recognize the stomach in the left example (which the non-stochastic method loops around), whereas this would not be easily recognizable if the image would be rotated.
>
> Regarding the “noise introduced by dying agents”: if dying agents over the entire length of the centerline would be excluded, the starting points of the segments in both directions would no longer be guaranteed to match up, as the median locations near the seed point would be re-computed from two different distributions. This can result in discontinuities near the seed points where the segments computed in both directions meet. Both problems can be fixed by simple smoothing methods, but we considered this outside of the scope of this work; we will address this in future work.
>
> We agree that arbitrarily defined “substantial” differences are too vague. We have now removed mentions of “substantial” differences and added statistical result comparisons using the Wilcoxon signed rank test (section 4.2).
>
> Regarding the term 4D MRI: thank you for pointing this out. We have changed it to 3D cine-MRI in the title and throughout the manuscript.
>
> The time dimension in the original images is indeed not aggregated as we would do in cardiac cine-MRI. To prevent confusion, we have updated the final paragraph of the introduction to mention the short time to scan each time-point.

---

### Official Review · AnonReviewer3 · 2021-03-05

**Confidence:** 4
**Preliminary Rating:** 3
**Recommendation:** Poster
**Final Rating:** 4

**Summary:**

The authors of the paper present two methods to extract the centreline of the bowel from cine-MRI. The target application is not a common area of research and the authors have clearly motivated and explained their intends. The authors propose both a deterministic and a stochastic method
The experimentation concerning the method is adequate but not extensive.

**Strengths:**

The paper is well written, both in terms of structure and in terms of language
The method is analysed in an adequate and easy to understand manner.
The method looks both interesting and powerful
There is an interesting and in depth discussion about limitations and abilities of the proposed methods

**Weaknesses:**

The method that the authors propose by their admittance is inspired by Wolternick et al. To what extend is this paper novel compared to the proposed solution of Wolternick et al ?

There is no comparison against other methods that perform centreline extraction, making the experimentation section of the paper lacking significantly.

**Deanonymize Review:**

no

**Final Rating Justification:**

I thank the authors for their response and answering my questions. I am upgrading my score to strong accept as I believe that the issues have been addressed to a satisfactory level. I believe the paper will benefit the community

**Justification Of The Preliminary Rating:**

Ive set my rating as a weak accept currently as it is not clear to what extend this method is novel compared to the aforementioned solution of Wolternick et al. A further reason is the lack of comparisons against other methods identified by the authors as related.
In general the paper is a good one , interesting and to the benefit of the community , if the aforementioned points are addressed then its a very strong one

Im willing to upgrade the rating given the authors provide sufficient argumentation in favour of the novelty  and comparative performance of their methods

**Paper Type:**

methodological development

**Questions To Address In The Rebuttal:**

To what extend is this paper novel compared to the proposed solution of Wolternick et al ?

**Special Issue:**

no

---

> ### Author Response · Authors · 2021-03-17
> **response to reviewer 3**
>
> We thank the reviewer for the comments; we have tried to address the concerns below.
>
> The innovation in this work is the stochastic tracking strategy. This strategy relies on a consensus from multiple agents as opposed to a single result. We add stochasticity to these agents in two ways. First, we randomize the starting position of each agent around the seed point. Second, we change the update procedure of each agent to be a stochastic process. Instead of choosing their next point based on the maximum network output, each agent chooses its next point by stochastically sampling a direction from the probability distribution predicted by the network (e.g. a direction with estimated probability 0.02 will have a 0.02 chance of being chosen by the agent). This adds additional randomness to the system and increases the variability in paths explored in the solution space. We have updated the methods section to better clarify the novelty (section 3.2, first paragraph).
>
> The main reason that we do not compare to other methods is that the only available similar work is intestinal centerline extraction in high resolution CT by Spuhler et al. 2006. Their method has only been evaluated in patients who were administered antispasmodic drugs in order to eliminate motion and achieve maximum distention of the intestines. This results in a much easier problem. As their method relies on the availability of a high-quality segmentation mask, it is not possible to apply this method to our data and compare the results. It may be possible to adapt methods from other centerline extraction problems for 3D cine-MRI of the small intestine, but doing so would likely require extensive development and this was considered outside of the scope of this work.

---

### Official Review · AnonReviewer4 · 2021-03-08

**Confidence:** 4
**Preliminary Rating:** 4
**Recommendation:** Oral
**Final Rating:** 4

**Summary:**

The authors proposed and evaluated a stochastic tracking approach for extracting centerlines of small intestines in 3D cine-MR images. The method is based on a 3D CNN orientation classifier and adopts a multi-agent consensus strategy to obtain centerline trajectories. In the experiments, the method was evaluated on 14 subjects with a leave-one-out manner and was shown to outperform a non-stochastic baseline approach in terms of better precision and sDSC scores. The authors also presented two cases qualitatively where the proposed stochastic method with multi-agent show better robustness in tracking than the non-stochastic one.

**Strengths:**

- The design of multiple stochastic agents reaching consensus makes the method less sensitive to local errors made by the CNN orientation classifier, which is a major advantage over the previous non-stochastic approach.
- The writing of the paper is very clear (the clinical problem, the challenges, the method, the experiments and results etc.), the structure is well organized, making reading the paper an pleasant experience.

**Weaknesses:**

- The major weakness of the proposed method is that it contains many hyperparameters that may require careful tuning. For example in the stochastic tracker, the number of agents, the radius around the seed point, the thresholding values for determining the stopping criteria, and also the distance threshold that was used for computing the evaluation metrics. How did the author tune these parameters? How sensitive are they to influence the tracking results? Have the authors done any analysis on these parameters?
- The proposed method was evaluated on a rather small dataset. Considering the variability in different subjects in this type of data, would the method with the same hyperparameter setting still work? If not, how easy would it be to tune those parameters?

**Deanonymize Review:**

no

**Detailed Comments:**

- In Table 1, there are also some segments where the non-stochastic method performs better than the stochastic approach. What could be the reason for that? Please elaborate this in the discussion section.
- In Figure 4 and 5, it might be helpful to provide some insights on the limitations of the proposed method if the authors could also visually show cases where the stochastic method is worse or failed. For example in the discussion, the authors wrote "... the stochastic method was more likely than the non-stochastic method to terminate early in regions with low contrast." This point could be illustrated with a few example images.
- How fast can the stochastic method run in general? Could the authors elaborate a bit on this?

**Final Rating Justification:**

The authors answered my questions clearly and well. I already voted for strong accept and I will maintain my rating to this paper.

**Justification Of The Preliminary Rating:**

The proposed method is solid, and could be interesting to a larger group of audiences. The validation of the method is overall informative and clear. The writing of the paper is excellent which makes the paper easy to follow.

**Paper Type:**

methodological development

**Questions To Address In The Rebuttal:**

It would be nice if the authors could address the questions in the weaknesses and detailed comments.

**Special Issue:**

yes

---

> ### Author Response · Authors · 2021-03-17
> **response to reviewer 4**
>
> We thank the reviewer for the helpful comments and suggestions.
>
> (w1): There are indeed a number of hyper-parameters in the method that could be tuned. To avoid overfitting to our (small) dataset, experimentally optimizing these hyper-parameters was kept to a minimum. Where possible, hyper-parameters were derived from physiological properties in the images: the initialization radius was chosen to match the thinnest non-contracted and non-collapsed regions, encouraging diversity without initializing agents in adjacent sections. The threshold for the confidence-based stopping criteria was determined in preliminary experiments using one image, by determining the network confidence when predicting directions for points along the manual centerlines, as well as for points along trajectories crossing the intestinal wall. The number of agents was chosen as a compromise between thorough exploration of the solution space and required computational resources; 32 agents was tried, but 64 performed better in the preliminary experiments. Originally, the image used in these experiments was included in the evaluation, but as multiple reviewers have pointed out, this allowed some information from our test set to leak into our hyper-parameter selection. We have now excluded this image, and updated the results in the paper to describe the 13 remaining folds (see: section 4).
>
> (w2): For images that are very different from any of the ones in our data set, the direction classifier might produce lower confidence estimates. In such cases, a lower threshold for the confidence-based stopping criteria would be desirable. Extensions of this work could investigate automatic re-calibration of these thresholds based on the input image, but such extensions would be outside of the scope of this work.
>
> (c1): We have extended the section discussing situations where the stochastic method may perform worse than the non-stochastic method (section 5, second paragraph).
>
> (c2): We completely agree with the reviewer that visual examples of such situations would be helpful; these have been added in the appendix.
>
> (c3): We have not focused on the efficiency of the implementation. The stochastic tracker with 64 agents runs at 2 iterations per second (=1mm/s with the settings used in the paper) on consumer hardware (a 3GHz CPU and a GTX2080). However, approximately 85% of that time is spent on sampling patches, as patch sampling is currently implemented sequentially on the CPU, resulting in slow performance. In future work, we plan to optimize the implementation (i.e. with asynchronous patch sampling on the GPU) and evaluate the computational performance.

---

> > ### Comment · AnonReviewer4 · 2021-03-20
> > **The distance threshold in evaluation**
> >
> > Thank the authors for the clear answers to my questions. I still have one minor remark about the hyperparameter choice. In the evaluation (section 3.3), the authors used a distance threshold to determine the correct tracking. This threshold was empirically set to 10 mm based on physiological property of the lumen. As this value directly influences the evaluation outcome, it would be great if the authors could do a sensitivity analysis on this threshold value if a future extended journal version of this paper is considered. For example, plot a figure to show how the change of this threshold value influences the recall, precision or sDSC.

---

### Official Review · AnonReviewer2 · 2021-03-08

**Confidence:** 3
**Preliminary Rating:** 2
**Final Rating:** 4

**Summary:**

This paper improves the centerline tracking algorithm of [Wolterink et al. 2019] by augmenting it with a stochastic tracking strategy. The algoritms works by considering _n_ stochastic agents and considers a majority voting strategy to pick the centerline. The method is validating using 15 scans (with leave-one out CV) and shows that stochastic method significantly improves from the non-stochastic original baseline.

**Strengths:**

1. The proposed solution makes sense and is a reliable extension. Considering the baseline picks the max of the posterior $p(D|x)$, it's a nice idea to extend the algorithm to include more randomness and increase the exploration to improve robustness of the results.


2. The paper is mostly well written. Notably, the introduction (Section 1.) is extensive, easy to read and presents excellently the state of the art and the motivation for the work.


3. The results corroborate the improvement of the algorithm, displaying enough significance to warrant using their solution rather than the original baseline.

**Weaknesses:**

1. Main issue relates to Section 3.2. This is the main novelty of the paper and I feel that it is badly presented. Judging from the text, there is not sufficient information that would allow a user to reproduce the algorithm. When using the tracking algoritm, is an image subvolume with the seed at the center used? The output of the original algorithm is a vector that maximises $p(D|x)$. This is just a direction given a subvolume and a previous centre point, how is the next voxel selected? Is the next subvolume in the image within the direction of the vector used? What do the authors mean when they stay state "the median location of all living agents is computed after each step"? Is the location of all living agents the majority voted voxel in the direction of the previous vector?


2. Unless I misunderstood, the "stochastic tracking strategy" is just the centerline tracking algorithm of Wolterink 2019 initiated at different seeds in a sphere with radius _r_ and using some stopping criterion based on the surviving agents. How is this different than just running the algorithm various times with different initialisations and computing $f(c_1, \dots, c_n)$ where $f(\cdot)$ is a function that returns the desired object. In my view, the proposal is too incremental and lacks novelty.

**Deanonymize Review:**

no

**Detailed Comments:**

1. What do the statistics look like if you considering recall/precision per scan for the entire centerline and hence compare results over 14 scans instead of 181 segments? Are the results still significant? Wouldn't it make sense to include this in the analysis considering you will use each computed centerline for patient specific analysis?


2. The proposed algorithm is an extension of the original tracking algorithm based on the pre-trained algorithm, which outputs candidate vectors. Thus, the method will always fail when the original algorithm fails. When does the original algorithm completely fail?


3. In Figure 3 - there are cases where the non-stochastic algorithm does better then the stochastic version. Why is this the case? Shouldn't the lower-bound in performance of the stochastic version be the non-stochastic result?


4. How is the initial seed obtained to start the tracking with the pre-trained network?


5. Could the network which generates candidate vectors be used in an autoregressive fashion somehow to improve the proposal of the vectors within a sphere?  Could the tracking algorithm be improved more significantly by extending the network that outputs the vectors?


6. Was 500 logits used to predict the vectors as in the original paper? Was this a hyperparameter determined by Wolterink on the dataset that was originally considered? Did the authors consider different numbers?


7. Why $n=64$ agents and $r=5$ mm? Was this determined on the same fold that the stopping criteria was selected on? If not, how can you be sure that you are not using the test set to pick hyperparameters? I understand there was insufficient data to perform hyperparameter selection using cross-val. However, it would be preferable to have some indication as to how crucial algorithm parameters affect performance.


8. I would personally move Section 2 Data to be in Section 4 Experiments and Results


9. Some further qualitative results in an Appendix would be welcome.

**Final Rating Justification:**

I would like to thank the authors for their extensive answers to my initial review. I am satisfied by their response and updates to the paper. I vote for acceptance of this paper as I feel now it would be beneficial to the community and that the novelty of the proposal method is now more apparent. Inclusion of new results also strengthens the manuscript. I am happy to move from "Weak Reject' to "Strong Accept".

**Justification Of The Preliminary Rating:**

My decision to give a preliminary rating of _Weak Reject_ is based on my main points made in _Weaknesses_. My overall impression is that the method is too incremental and does not present much novelty. Moreover, I find the main section of the paper (that explains the novelty) poorly explained. Some small other points in _Detailed Comments_ which can be addressed to improve over all the quality of the manuscript.

I am happy to be challenged on main points about novelty of the method. I would be happy to raise the score towards _Accept_ if I can be convinced about the significance of the proposal.

**Paper Type:**

methodological development

**Questions To Address In The Rebuttal:**

Main question to address which would change my mind would be based on the main points of Weaknesses. Other small points to address in _Detailed Comments_ would be welcome too.

**Special Issue:**

no

---

> ### Author Response · Authors · 2021-03-17
> **response to reviewer 2**
>
> We thank the reviewer for the helpful comments and suggestions.
>
> (w1): We recognize that Section 3.2 in the original manuscript sacrificed clarity for brevity. We have updated the section to include more details. Specifically, there might be some misunderstanding regarding the behavior of agents. Our method performs the following: For each agent at each timestep, a patch is sampled with the agent at its center. This patch is subsequently passed to the neural network and a next point is predicted, resulting in a new centerline point for this agent. The median location, i.e. next centerline point is the median of all of these agents’ coordinates. Hence, the next predicted intestine centerline point is implicitly inferred from the current points of all living agents. In the next step, agents are not re-initialized; the progress of the agents is independent from the predicted intestine centerline point apart from the aberrant termination condition. This has been added to section 3.2.
>
> (w2): While the agents behave similarly to the iterative tracker proposed by Wolterink et al., there is an important difference: The agents do not choose their next point based on the maximum network output, but instead choose their next point by stochastically sampling a direction from this distribution (e.g. a direction with estimated probability 0.02 will have a 0.02 chance of being chosen by the agent). This adds additional randomness to the system and increases the variability in paths explored in the solution space, improving robustness.  We have updated the methods section to better clarify this novelty (section 3.2, first paragraph).
>
> (c1): We thank the reviewer for the suggestion regarding the pooling of the results on a patient level. Analysis was done on the segment level as these are sections of visibly connected anatomy, but we agree that analyzing results pooled on the patient level is valuable. This has been added to the result section of the manuscript and extended Figure 3.
>
> (c2): Examples of cases where the network fails are in ringing artifacts, air bubbles or blurry regions at the very edge of the FOV. Such examples have been added to the appendix.
>
> (c3): While the stochastic method prevents “unlucky” outliers, it also prevents “lucky” ones: if a large number of agents is terminated in an area with artifacts or low contrast (either due to low network confidence or disagreement between agents), tracking terminates. However, a single non-stochastic tracker may get lucky and make it through such an area by virtue of having fewer stopping conditions, which can lead to a much greater recall. As can be seen in figure 3, this does not occur frequently. We have slightly expanded the discussion on these situations (section 5, second paragraph) and visual examples where this does happen have been added in the appendix.
>
> (c4): The initial seeds are considered input to the method. These can either be obtained by clicking an intestinal section in an interactive interface or by an automatic seed selection method. In this work, the initial seed is sampled from the manual centerline (section 4.1, last sentence).
>
> (c5): We thank the reviewer for the suggestion. As decisions in subsequent timesteps are correlated, an autoregressive component could indeed improve performance. Improvements to the direction proposal network were out of scope for this work, but will definitely be considered in future work.
>
> (c6): Indeed, the number of logits was kept the same as in the prior work. We tried to keep hyper-parameter optimization to a minimum to avoid overfitting to our small dataset and Wolterink et al. showed their network was not very sensitive to this hyper-parameter (figure 6 in their paper).
>
> (c7): Choice on the initialization radius (r=5mm) followed from the physical properties of the intestines: This is approximately the minimum radius for non-contracted non-collapsed intestinal sections, encouraging diversity without initializing agents in adjacent sections. The number of agents was chosen as a compromise between thorough exploration of the solution space and required computational resources, but this hyper-parameter was not extensively optimized. In the preliminary experiments using one image, n=64 performed slightly better than n=32, which was the only other value that was investigated. As multiple reviewers have pointed out, the preliminary experiments allowed some information about the test set to flow into the hyper-parameter selection. We have excluded the image used in these experiments and re-run the result analysis. The manuscript is updated to reflect the numbers from the remaining 13 folds.
>
> (c8): We agree this might be a good option, but given the space limitations and limited time, we would for now prefer to keep the current structure.
>
> (c9): We have provided additional visual results in the Appendix.

---

### Meta-Review · Area_Chair1 · 2021-03-29

**Recommendation:** Accept (Poster)

**Metareview:**

The paper presents a method to extract intestine centerline in 3D cine-MRI. The methodological contribution lies in adding stochasticity to an existing centerline tracking method by establishing a consensus among the multiple stochastic agents.  R1, R2, and R3 rated the methodological contribution as small or incremental. Although strongly building on an existing prior work, there is some novelty in the methodological aspects linked to the stochasticity, as recognized by R3 after the rebuttal, and in the methodology, as mentioned by R1. R2 and R1 raised some issues regarding the clarity of the methodology and requested further analysis of the results. Most of these questions were addressed satisfactorily in the rebuttal and revised version. R2 and R4 both raised some concerns regarding the hyperparameter tuning and setting, although the rebuttal claims to keep experimental optimization of the hyperparameters to a minimum relies instead on physiological prior knowledge, authors might want to look at a more in-depth analysis of their sensitivity and influence in the future. The same goes for considering other baseline methods, a larger database, and different evaluation metrics.  After the rebuttal, there was however a consensus among the reviewers on the value of publishing this work in MIDL in its current state.

**Paper Type:**

both

---

### Decision · Program_Chairs · 2021-03-31

**Decision:**

Accept

**Comment:**

Congratulations your paper has been selected as a long oral.